# Prognostic Significance of Cuproptosis-Related Gene Signatures in Breast Cancer Based on Transcriptomic Data Analysis

**DOI:** 10.3390/cancers14235771

**Published:** 2022-11-24

**Authors:** Zizhen Zhou, Jinhai Deng, Teng Pan, Zhengjie Zhu, Xiulan Zhou, Chunxin Lv, Huanxin Li, Weixiong Peng, Bihai Lin, Cuidan Cai, Huijuan Wang, Yufeng Cai, Fengxiang Wei, Guanglin Zhou

**Affiliations:** 1Department of Breast Surgery, Longgang District Maternity& Child Healthcare Hospital, Shenzhen 518172, China; 2Richard Dimbleby Laboratory of Cancer Research, School of Cancer & Pharmaceutical Sciences, King’s College London, London SE1 1UL, UK; 3Xiangjiang Laboratory, Changsha 410205, China; 4Oncology Department, Punan Hospital of Pudong New District, Shanghai 200125, China; 5Nanomaterials and Spectroscopy Group, Electrical Engineering Division, Cambridge CB3 0FA, UK; 6Hunan Zixing Intelligent Medical Technology Co., Ltd., Changsha 410221, China; 7Department of Colorectal Surgery, Sir Run Run Shaw Hospital, Zhejiang University School of Medicine, Hangzhou 310058, China; 8Genetics Lab, Longgang District Maternity & Child Healthcare Hospital, Shenzhen 518172, China

**Keywords:** cuproptosis, tumor immune microenvironment, prognostic model, breast carcinoma

## Abstract

**Simple Summary:**

Breast cancer, as the leading cause of cancer-related deaths in women, still poses a lethal threat to human health worldwide. To understand the involvement of cuproptosis, a new version of cell death, in the prediction of prognosis of breast cancer patients, we built a nomogram model based on the differentially expressed cuproptosis-related genes, finding out that the cuproptosis-related signature is useful for stratifying patient subtypes and is closely related to the tumor immune microenvironment.

**Abstract:**

Breast cancer (BRCA) remains a serious threat to women’s health, with the rapidly increasing morbidity and mortality being possibly due to a lack of a sophisticated classification system. To date, no reliable biomarker is available to predict prognosis. Cuproptosis has been recently identified as a new form of programmed cell death, characterized by the accumulation of copper in cells. However, little is known about the role of cuproptosis in breast cancer. In this study, a cuproptosis-related genes (CRGs) risk model was constructed, based on transcriptomic data with corresponding clinical information relating to breast cancer obtained from both the TCGA and GEO databases, to assess the prognosis of breast cancer by comprehensive bioinformatics analyses. The CRGs risk model was constructed and validated based on the expression of four genes (NLRP3, LIPT1, PDHA1 and DLST). BRCA patients were then divided into two subtypes according to the CRGs risk model. Furthermore, our analyses revealed that the application of this risk model was significantly associated with clinical outcome, immune infiltrates and tumor mutation burden (TMB) in breast cancer patients. Additionally, a new clinical nomogram model based on risk score was established and showed great performance in overall survival (OS) prediction, confirming the potential clinical significance of the CRGs risk model. Collectively, our findings revealed that the CRGs risk model can be a useful tool to stratify subtypes and that the cuproptosis-related signature plays an important role in predicting prognosis in BRCA patients.

## 1. Introduction

Breast cancer is a serious threat to women’s health worldwide, and the incidence and mortality rates have not declined during recent years [1]. In accordance with epidemiological data, by 2030, the number of women diagnosed with breast cancer will approximately double to 3.2 million [2]. The current treatment regimens for BRCA mainly include chemotherapy, radiotherapy and radical surgical therapy. However, due to different mechanisms of origin of tumors and the lack of response to chemotherapy drugs in some cases [3], cancer recurrence and metastasis often tend to occur. Hence, to improve the prognoses of breast cancer, more effective personalized therapies need to be developed.

Copper is an indispensable element for the biological activities of cells in organisms, and unbalanced copper in tissue leads to the occurrence of various diseases [4]. Copper has been reported to induce cell death through a new cell death mechanism, coined as cuproptosis [5], which is defined as the presence of copper accumulation and lipoylated TCA cycle proteins and is distinct from apoptosis [6], ferroptosis [7] and pyroptosis [8]. Specifically, cuproptosis acts through protein lipidation in the TCA cycle [5], where copper binds to the protein fatty acyl moiety, leading to the aggregation of protein, the loss of Fe-S cluster proteins and cellular death due to toxic stress [5]. Therefore, cuproptosis-mediated cytotoxicity could be a novel direction for breast cancer management. However, the role of cuproptosis-related genes in BRCA is still elusive.

In this study, we firstly identified cuproptosis-related genes (CRGs) with differential expression between breast cancer and normal tissue. Then, a risk model based on the CRGs was constructed to predict prognosis. Additionally, we showed the association between the CRGs score and somatic mutations/immune cell infiltration level/tumor mutation burden. Collectively, we hope to find potential prognostic biomarkers for BRCA and to lay the foundation for new therapeutic targets.

## 2. Materials and Methods

### 2.1. Data Download and Preprocessing

We obtained the breast cancer gene expression dataset and corresponding information (FPKM standard) from the TCGA database (https://portal.gdc.cancer.gov/, accessed on 1 July 2022), with 103 normal tissues and 1109 tumor tissues. The RNA-seq data and clinical information were retrieved from the GEO database (https://www.ncbi.nlm.nih.gov/geo/, D: GSE21653, accessed on 18 July 2022) as a validation dataset. Gene ID was converted using perl software( http://www.perl.org/, Version 3.8, Larry Wall, Los Angeles, CA, USA). Somatic mutation data and copy number alteration information were also downloaded from the TCGA database.

### 2.2. Acquisition of Differentially Expressed Genes Related to Cuproptosis

We identified the differentially expressed genes (DEGs) of cuproptosis-related genes between normal tissues and BRCA patient tissues in TCGA by using the “limma” package. Then, we used STRING software (https://string-db.org/, version 11.5, Damian Szklarczyk, Zurich, Switzerland) to construct a protein interaction network of different cuproptosis-related genes and to analyze the correlation of them.

### 2.3. Identification of Breast Cancer Subtypes Based on Different Cuproptosis-Related Genes

Consensus clustering is an algorithm used to verify the rationality of clustering [9]. We constructed a consensus matrix using the “ConsensusClusterPlus” package in the R software, identifying BRCA subtypes based on differential CRGs between normal tissue control and breast cancer patients. Further, we analyzed the clinical relevance between the two subtypes, including the clinical parameters and overall survival (OS).

### 2.4. Identification Differential Expression Gene and GSEA Functional Analysis between the Subtypes

DEGs were screened and selected between the two subtypes using the “limma” package in R software (*p* < 0.05, Foldchange = 2). GSEA analysis was carried out by using the “enrichment plot” package and “clusterProfiler” packages on the basis of the human gene annotation set, which is derived from the MSigDB database, to determine biological signaling. 

### 2.5. Construction and Validation of the CRGs Prognostic Model

Firstly, we selected candidate CRGs in the training set (TCGA-BRCA cohort) by univariate and multivariate cox regression analysis using the “limma” package. Four genes related to overall survival (OS) were filtered and further used for lasso regression analysis to reduce overfitting of the model. We calculated the correlation coefficients of candidate genes in the “glmnet” package and used it to construct a prognostic model for breast cancer patients. The formula for the gene model was as follows [10]: gene risk score = Σ(Exp*Coef). The subject operating curve (ROC) was used to evaluate the effectiveness of the model.

### 2.6. Estimation of Immune Cell Infiltration Based on the CRGs Model

A single-sample gene set enrichment analysis (ssGSEA) [11], calculating enrichment scores based on sample and gene set pairings, was performed to estimate immune cell infiltration. The relationships between risk score and immune cells/immune signaling pathways were also examined. Additionally, we grouped tumor mutational burden according to the risk group and performed survival analyses between high and low tumor burden subgroups.

### 2.7. Construction of Disease Prediction Models and Screening of Independent Prognostic Factors among the CRGs Risk Score and Clinical Indicators

Univariate analysis and multivariate analysis were performed to screen for independent prognostic factors. Based on these factors, the predictive clinical nomogram model was then constructed. The model calibration curve was further analyzed to compare the bias between the predicted OS value and the actual OS value. The subject operating curve (ROC) was used to evaluate the effectiveness of the model. Decision curve analysis [12] (DCA) was used to evaluate the clinical application value of the model.

### 2.8. Statistical Analyses

All the analyses were conducted in R software (https://cran.r-project.org/src/base/R-4/, Version 4.0.4, Ross Ihaka and Robert Gentleman, Auckland, New Zealand). Perl software (http://www.perl.org/, Version 3.8, Larry Wall, Los Angeles, CA, USA) was used for Gene id text conversion. Categorical variables and continuous variables were analyzed using Pearson’s chi-square test and one-way ANOVA, respectively, with a *p* value < 0.05 being significant. Kaplan–Meier analysis was performed to analyze the overall survival of breast cancer patients, and then, univariate and multivariate Cox regression analyses were performed to select the significantly differential variables (including independent prognostic factors and clinical features). The Mann–Whitney U test was performed to analyze the differences in immune cell infiltration. An overview of the data analysis process is shown in Appendix A.

## 3. Results

### 3.1. Differential CRGs Expression between Para-Carcinoma Tissues and Tumor Tissues

In total, we retrieved 19 cuproptosis-related genes (CRGs) [5,13,14,15,16] based on published studies (including NFE2L2, NLRP3, LIAS, LIPT1, ATP7B, ATP7A, SLC31A1, FDX1, LIPT2, DLD, DLAT, CDKN2A, DBT, MTF1, GLS, GCSH, DLST, PDHA1 and PDHB) and found that 18 CRGs were differentially expressed (such as NFE2L2, NLRP3, LIAS, LIPT1, ATP7B, ATP7A, SLC31A1, FDX1, DLD, DLAT, CDKN2A, DBT, MTF1, GLS, GCSH, DLST, PDHA1 and PDHB) when comparing the gene expression between 1109 breast cancer tissues and 113 para-carcinoma tissues (Figure 1A,B). The mRNA levels of CRGs are shown in both a heat map (Figure 1A) and histograms (Figure 1B). Among them, the expression of the majority of genes was decreased in tumor samples except for four genes (ATP7B, SLC31A1, PDHB and CDKN2A), suggesting cuproptosis signaling was significantly downregulated in breast tumors. To explore the interactions among expression-differential CRGs, a protein–protein interaction (PPI) analysis was performed, and the result was illustrated with the minimum interaction score of PPI analysis set at 0.9 (Figure 1C). Importantly, we found that LIAS, GCSH, DLD, DBT, LIPT, PDHB, DLAT, DLST and PDHA1 were hub genes (Figure 1C). Moreover, the correlation analysis among different CRGs was further visualized (Figure 1D). The results showed that most of the cuproptosis-related genes were positively correlated. Notably, ATP7B was observed to be negatively correlated with multiple genes (PDHA1, CDKN2A and GCSH).

### 3.2. Identifying Tumor Subtypes Based on Differentially Expressed CRGs

As our results suggested the close involvement of cuproptosis signaling in breast cancer, to further explore the impact of CRGs on breast cancer stratification, an unsupervised consensus clustering analysis was performed on 1109 tumor patients in the TCGA cohort. The cluster variable range was set from 2 to 9, with the calculated optimal cluster number (K = 2) showing the highest intragroup correlations and the lowest intergroup correlations (Figure 2A–D and Appendix A). Therefore, these data suggested that 1109 breast cancer patients could be sub-grouped into two clusters based on CRGs (namely C1 and C2). Then, we correlated sub-clusters with differential expression of genes and clinical parameters including survival and other clinical characteristics (age, gender, T stage, N stage, M stage, tumor stage) (Figure 2E,F). The results showed that there was a significant difference between the two groups regarding tumor stage, T stage and survival (Figure 2E,F). Specifically, breast cancer patients of cluster 1 showed an improved clinical outcome (Figure 2F).

To further understand the associated functional signaling pathways, GSEA analysis revealed that cluster C1 showed an enrichment in innate immune activation pathways, such as the cytosolic DNA sensing pathway and natural killer cell mediated cytotoxicity pathway (Figure 3A). However, cluster 2 was enriched in the inflammation pathway, including JAK-STAT signaling (Figure 3B). These data suggested that the activation of innate immune signaling and the enhancement of NK cell cytotoxicity favor patient prognosis. Moreover, we found eight cuproptosis related genes differentially expressed between these two subtypes (Figure 3C).

### 3.3. Establishment of a CRGs Risk Model

As previous results have shown the important impacts of cuproptosis on breast cancer, we hypothesized that a risk score model based on cuproptosis genes provides an opportunity to predict BRCA prognosis. Univariate cox analysis and muticariate cox analysis were then performed and identified that four genes were independent prognostic factors, namely NLRP3, LIPT1, PDHA1 and DLST (Figure 4A). Among them, the hazard ratio (HR) of NLRP3 and LIPT1 was less than 1, indicating a protective impact, while the HR of PDHA1 and DLST was greater than 1, showing risk effects. To further reduce model overfitting, LASSO regression analysis was used to select optimal gene variables, and four gene signatures were also chosen according to the optimal λ value (Figure 4B,C). The formula for calculating risk scores was as follows: (−0.499 × NLRP3 exp.) + (−0.340 × LIPT1 exp.) + (0.062 × PDHA1 exp.) + (0.0714 × DLST exp.). Based on the median risk score, we divided 914 breast patients into low- and high-risk subgroups. PCA and tSNE analyses showed that the two clusters were well separated (Figure 4D,E). Compared with the low-risk group, the high-risk group experienced a higher death rate and shorter overall survival (Figure 4G–I), which was further verified in the validation cohort set (Figure 5A–F). The area under the ROC curve in both the training and validation sets at 1, 3 and 5 years was (AUC) 0.686, 0.722 and 0.727, and 0.626, 0.663 and 0.678, respectively (Figure 4F and Figure 5C). In addition, we compared our risk model with other established models and found that the AUC of our risk model was higher than that of others (Appendix A) [17,18]. 

### 3.4. Somatic Mutation and Functional Analysis Based on Risk Subgroups

We consistently compared the transcripts of genes between high- and low-risk groups and observed that the high-risk group showed a higher expression in PDHA1 and DLST, but a lower expression in NLRP3 and LIPT1 (Figure 6A). Further, we analyzed the 15 most common gene mutations in the high- and low-risk groups and found that PIK3CA mutation rates were higher in the high-risk group than in the low-risk group, while TP53 showed the opposite trend (Figure 6B,C). These data indicated that the high-risk group was associated with an oncogene mutation, but the low-risk group was associated with a tumor suppressor gene mutation.

We further explored the enrichment of molecular biological functions between the high- and low-risk groups by both GO and KEGG analyses (Figure 7A,B and Appendix A). The GO analysis showed that the high-risk group was mainly enriched in signaling pathways including “focal adhesion”, “cell-substrate junction” and “cadherin binding”, with low risk in “collagen-containing extracellular matrix” and “extracellular matrix organization” (Figure 7A and Appendix A). Moreover, KEGG analysis identified that the high-risk group was enriched in “protein processing in endoplasmic reticulum”, “cell cycle” “carbon metabolism” and “cellular senescence”, but the low-risk group showed more abundance in “PI3K-Akt signaling pathway” and “proteoglycans in cancers” (Figure 7B and Appendix A).

### 3.5. Immune Infiltration, Immune Evasion and Tumor Mutational Burden Analysis Based on the Riskscore Subgroups

Next, we sought to understand different immune phenotypes between the high- and low-risk groups. The ssGSEA analysis was performed to estimate the enrichment scores of 16 immune cell types and 13 immune-related pathways between these two subgroups in both TCGA-BRCA and GEO datasets (Figure 8A–D). The results showed that the immune infiltrates in the high-risk group were generally higher than in the low-risk group, especially for aDCs, DCs, macrophages, Tfh, Th1 cells, Th2 cells and Tregs (Figure 8A,C). In line with this, a similar trend was observed with the high-risk group associated with a higher degree of immune pathway activation (Figure 8B,D). Further, we performed correlation analyses between the CRGs risk score and the immune signaling pathways. The results showed that immune signaling pathways were positively intercorrelated (Appendix A). However, the CRGs risk scores were shown to be negatively correlated with diverse immune cells (Appendix A) and various immune signaling pathways (Appendix A). Moreover, the high-risk group was observed to be more prone to immune escape (Figure 8E). These data indicated that the stratification of clusters was associated with immune regulation. When analyzing the tumor mutational burden (TMB), the results showed that the high-TMB group was associated with a poor prognosis (Figure 8F). When combined with TMB, the low-risk group with a low TMB burden presented the best prognosis (Figure 8G).

### 3.6. Independent Factor Prognostic Value of the Risk Model and Construction of the Nomogram

To further investigate the clinical significance of the CRGs risk score model, we combined the risk model with clinical indicators to construct a clinical prediction model. Firstly, univariate and multivariate cox regression analyses were used to assess the value of the risk score as an independent prognostic factor. Specifically, univariate cox regression analysis showed that risk score was an independent factor for OS prediction in both the TCGA and GEO datasets, with HR = 2.806 (95% CI: 1.491–5.280) and 2.848 (95% CI: 1.802–4.502), respectively (Appendix A). Moreover, multivariate regression analysis confirmed that the risk score was still an independent prognostic factor in both cohorts after adjustment for other confounding factors (Appendix A). Then, we constructed a clinical nomogram model with the TCGA dataset based on clinical indicators and risk scores (Appendix A), and the calibration curves showed that the predicted cure at 1, 3 and 5 years had a small deviation compared with the actual curve (dotted line), indicating that risk score was a good scoring value (Appendix A). In addition, in order to verify the accuracy of our nomogram model, we compared the AUC values between our model and the TNM model, and the results showed that the AUC values of our model at 1, 3 and 5 years were 0.825, 0.771 and 0.748, respectively, which were higher than that of the TNM model (Appendix A)). In addition, we used decision curve analysis (DCA) to analyze the clinical application value of our model. Similarly, the values of the area under the DCA curve of our model were higher than those of the TNM model (Appendix A).

## 4. Discussion

Globally, breast cancer is one of the most common malignant tumors, posing a serious threat to women’s health. However, the commonly used treatments in clinic [2] cannot effectively reduce patient mortality rates due to the heterogenous responses of tumors and even drug resistance [19]. Therefore, stratifying the potential subpopulation of patients is essential for the development of individual treatment, which is beneficial for patients’ clinical outcomes.

Copper, an important enzymatic cofactor, mediates many physiological functions of cells. However, dysregulation of intracellular copper stores leads to oxidative stress and even cytotoxicity [20]. Current studies have revealed that copper is closely involved in cell proliferation, tumor metastasis, angiogenesis and the remodeling of tumor microenvironments via various molecular mechanisms [21,22]. For example, copper complexes have been observed to be involved in ferroptosis of pancreatic cancer cells [23]. Additionally, an intracellular copper delivery system [24], modulated by the ATP7A-LOX pathway, can promote tumor growth and metastasis in breast cancer. Thus, these observations underline the important role of copper in different types of cancer. Recently, the accumulation of copper has been discovered to trigger a new version of cell death, cuproptosis, involving protein fatty acylation [5]. To be specific, excessive deposition of intracellular copper can induce fatty acylated dihydrofatamide s-Acetyltransferase (DLAT) aggregates and affects the TCA cycle, resulting in proteotoxic stress and, consequently, cell death [13]. Moreover, since other studies have suggested the close relationship between cuproptosis and breast cancer [25], we speculate that the induction of cuproptosis in breast carcinomas may become a new avenue for treating breast cancer.

In this study, we established a cuproptosis-related gene (CRG) signature based on the differentially expressed genes between non-breast cancer and breast cancer patient samples. According to the expression level of CRGs, patients were grouped into two clusters (C1 and C2), with C1 being positively associated with innate immune activation (especially NK cell cytotoxicity) and a favorable prognosis. Further, univariate and multivariate analyses were performed and four candidate genes (NLRP3, LIPT1, PDHA1 and DLST) were further selected to build a risk model. Accordingly, the low-risk group had a more favorable overall survival. NLRP3 belongs to the NLR family of pattern recognition receptors, which are important in innate immunity [26,27,28], including the modulation of the polarization of tumor-associated macrophages to M1 [29]. Moreover, NLRP3 mediates pyroptosis by triggering caspase 8 activation [30], indicating its roles in regulating tumor cell death and tumor–immune interaction. Similar to other studies [30,31], our results showed that NLRP3 was decreased in breast cancer tissues and was associated with a favorable prognosis. LIPT1 (lipoyltransferase 1) is involved in the TCA cycle and transfers essential lipoic acid cofactors to mitochondrial 2-keto acid dehydrogenase [5]. Previous studies have shown that LIPT1 is positively correlated with tumor progression [32]. PDHA1, the E1 subunit of PDHc, is responsible for activating glycolysis and tricarboxylic acid synthesis [5]. PDHA1 has been shown to be either beneficial or detrimental in a cancer type-specific manner [33,34,35]. Consistent with other studies [35], our analysis indicated that PDHA1 is an independent prognostic factor for OS and has a positive correlation with histological type and tumor size. Finally, DLST is closely involved in cell metabolism, contributing to the transduction of α-ketoglutarate into succinyl-CoA10 in the TCA cycle in an irreversible manner [36]. In several cancer types, DLST is found to be negatively correlated with clinical outcomes [37]. Collectively, candidate genes, used to establish the risk model, have already been identified as being closely involved in different aspects of tumor behavior and patient prognosis.

Cancer has been widely considered to be the result of the accumulation of genetic mutations, including the gain of function of oncogenes and loss of function of tumor suppressors [38,39,40]. We analyzed 15 common somatic gene mutation rates between the high-risk and low-risk subgroups and found significant differences in P53 and PIK3CA mutations, with the high-risk group being positively associated with a PIK3CA mutation and the low-risk group being positively associated with a p53 mutation. Immune infiltration and immune scores have been identified to be critical in the tumor microenvironment and patient prognosis [41,42]. There is also evidence that copper-related carriers are associated with tumor immunity, which affects the prognosis of the disease [13,32,43], emphasizing the relationship between the copper-regulatory system and immune regulation. Here, our study revealed that breast cancer patients in the high-risk group generally showed higher immune infiltration and greater enrichment of immune-related pathways than their counterparts in the low-risk group, indicating that the CRGs risk score model can distinguish the degree of immune infiltration in patients, and cancer-associated inflammation may play important roles in patient prognoses [44]. Of note, our data also suggested that NK cell activation may help to stratify subpopulation of patients, which makes it a promising new target for tumor immunotherapy [45]. Additionally, the comparison regarding immune escape between the high- and low-risk subgroups revealed that the high-risk group was associated with a higher potential for immune escape. Furthermore, the tumor mutational burden (TMB) analysis showed that the patients in the high-risk group were generally associated with a poorer prognosis independent of TMB status, and the subtype of patients in the low-risk group with a low tumor mutational burden showed the most favorable prognosis. Finally, we constructed a clinical prognostic nomogram model based on CRGs in breast cancer patients, which was able to accurately predict clinical outcomes. Therefore, our data suggest that regulating the level of copper in the body is a novel direction for the management of cancer [46].

Our study also showed some limitations. Firstly, Jia li et al. [18] recently reported a model based on 13 differentially expressed cuproptosis regulators (between breast cancer and normal samples) selected from the original report firstly discovering cuproptosis [5], which could efficiently predict prognosis and even response to immunotherapy. Our study applied expanded cuproptosis-related DEGs according to different studies [5,13,14,15,16]. Therefore, our established CRGs risk model showed higher AUC values in predicting prognosis. However, more studies have been contributing to finding additional relevant genes. Thus, more potential candidates could be available in the future for better predictive model establishment. Importantly, it is impossible to reveal heterogeneity with only the factors associated with cuproptosis signaling. It is hoped that we can provide proof of concept that cuproptosis signaling may be of great potential as part of the biomarker panel. In future research, we hope to combine cuproptosis-associated genes with other essential molecularly and clinically independent prognostic factors to establish more sophisticated risk models for improved patient stratification.

Taken as a whole, our study showed that cuproptosis-related genes are closely correlated with breast cancer, which are differentially expressed between normal and breast cancer tissues. Furthermore, in both the validation set and training set, a risk score model was generated based on the risk gene signature of the four CRGs, which is capable of independently predicting the prognosis of BRCA. Additionally, the risk subgroups based on risk scores were closely related to tumor immunity. Collectively, our research developed a novel gene signature to predict the prognosis of BRCA and to offer a solid foundation for further research on the relationship between CRGs and the tumor immune microenvironment.

## 5. Conclusions

Among the cuproptosis hub genes, four genes were identified as independent prognostic factors—NLRP3, LIPT1, PDHA1 and DLST—and they were used to build a nomogram model. Based on this model, patients were classified into high-risk and low-risk subgroups. Our data showed that the high-risk group was associated with an oncogene mutation, but the low-risk group was related to a tumor suppressor gene mutation. Importantly, the high-risk group was more prone to immune escape and was associated with a poor prognosis.

## Figures and Tables

**Figure 1 cancers-14-05771-f001:**
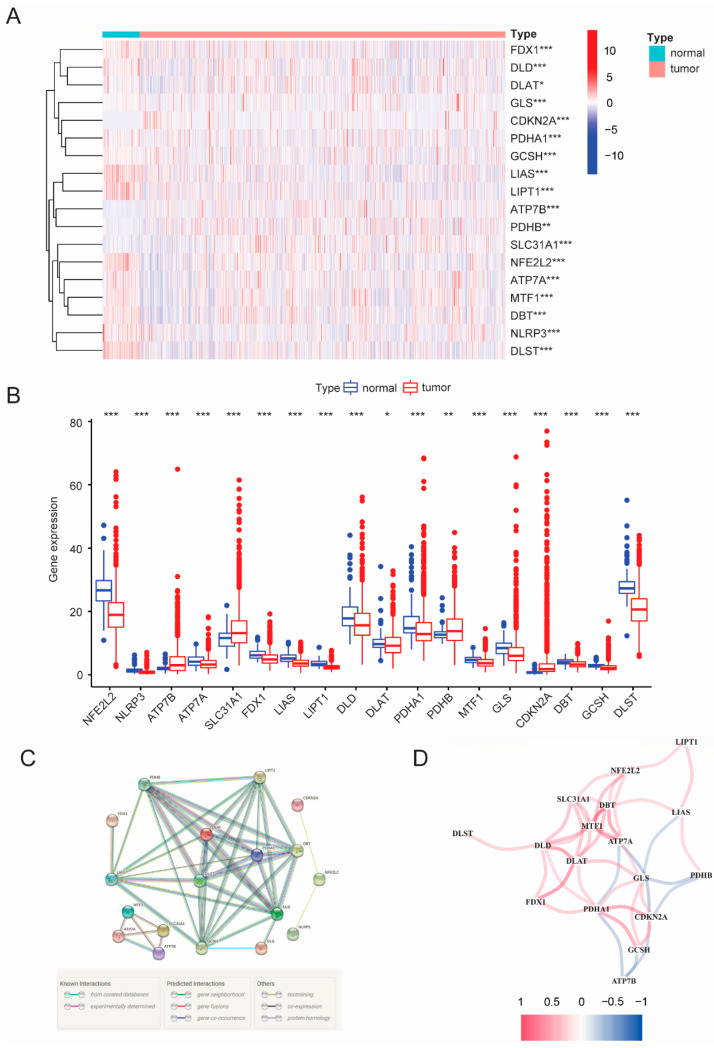
Expressions and interaction relationship of the 19 CRGs between breast para-carcinoma tissues and tumor tissues. (**A**) Heatmap of the CRGs between the para-tumor and the tumor tissues (red: high expression gene, blue: low expression gene). The *p* values were set as: * *p* < 0.05; ** *p* < 0.01; *** *p* < 0.001. (**B**) Bar diagram of the comparison between para-tumor and tumor tissues regarding expression of 19 CRGs. (**C**) PPI network of the CRGs. (**D**) The correlation network of the 19 CRGs (red: positive correlation; blue: negative correlation; color depth represents correlation strength).

**Figure 2 cancers-14-05771-f002:**
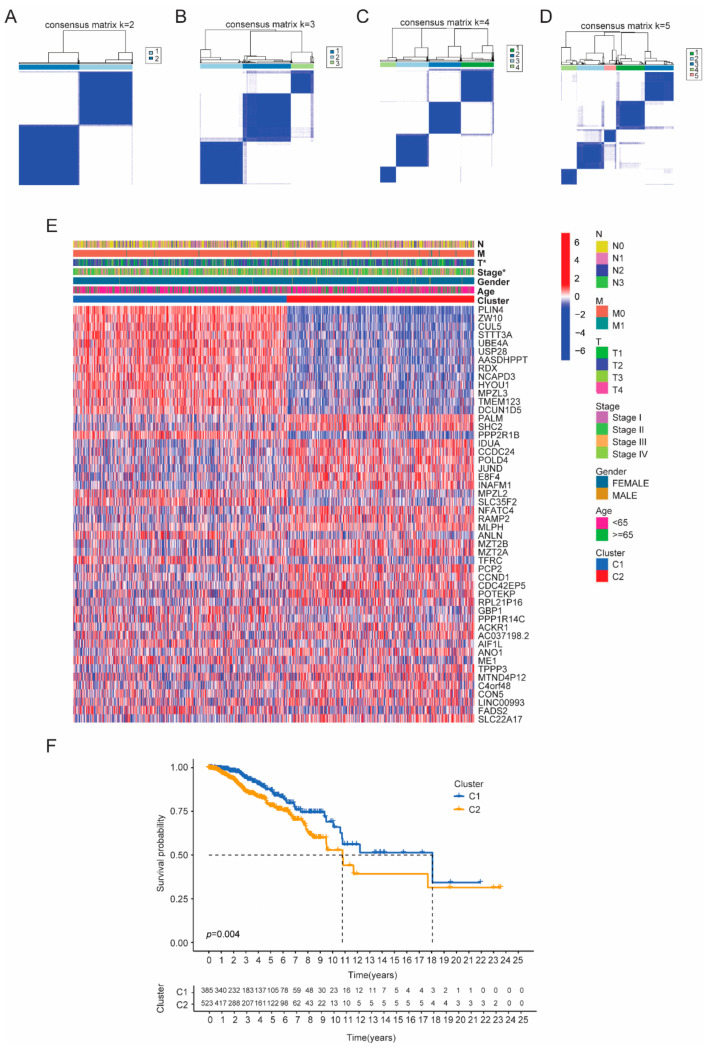
Identification of cancer patients subclusters based on the CRGs. (**A**–**D**) A total of 1109 BRCA patients were grouped into two clusters according to the consensus clustering matrix (k was set as 2–5). (**E**) Heatmap of the correlation between subclusters and DEGs or the clinicopathologic characters. (**F**) Kaplan–Meier OS curves analysis for cancer patients according to the two clusters.

**Figure 3 cancers-14-05771-f003:**
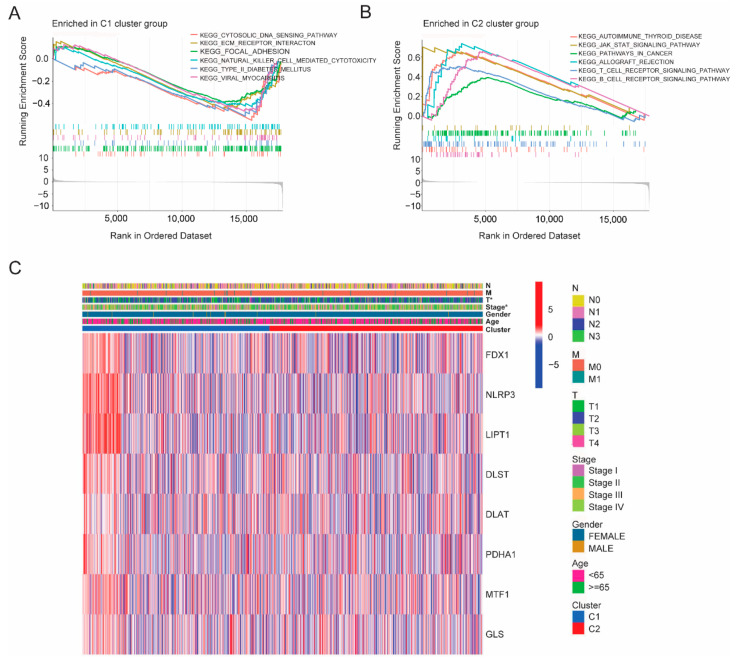
GSEA and the cuproptosis-related DEGs based on the breast cancer clusters. (**A**) GSEA analysis of DEGs based on cluster group 1 (C1). (**B**) GSEA analysis of DEGs based on cluster group 2 (C2). (**C**) A heatmap of cuproptosis-related DEGs based on the breast cancer clusters.

**Figure 4 cancers-14-05771-f004:**
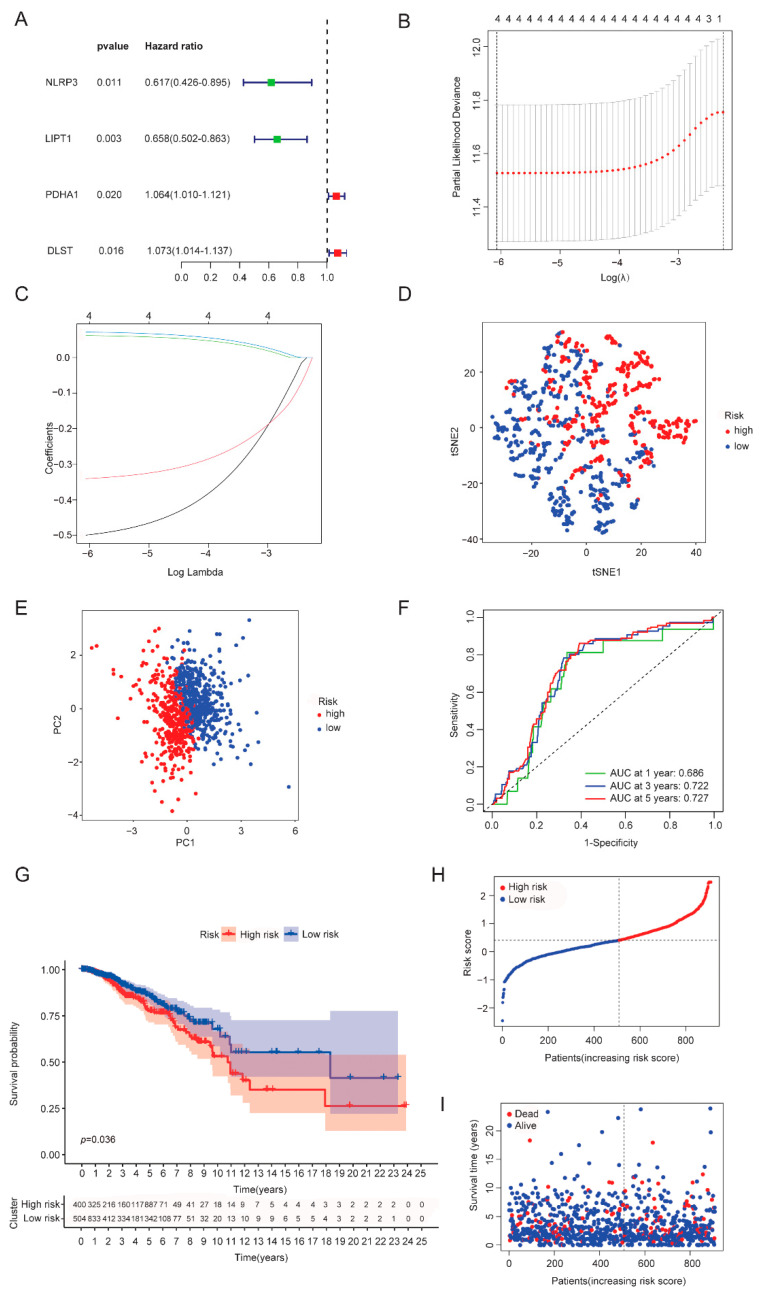
Identification of the CRGs risk model in the training set (TCGA cohort). (**A**) Forest plot of multi-variate cox regression analysis for each cuproptosis gene, and four candidate genes with *p* < 0.05. (**B**) Partial likelihood deviance plot of LASSO regression for four candidate genes. (**C**) Cross-validation for four OS-related genes in the LASSO regression (Different color lines represent different genes: black line (NLRP3), red line (LIPT1), green line(PDHA1) and blue line(DLST). (**D**) tSNE plot for BRCA patients based on the risk score. (**E**) PCA plot for BRCA patients based on the risk score. (**F**) ROC curve analysis demonstrated the predictive efficiency of the risk score. (**G**) Kaplan–Meier curve analysis for the OS of patients in the two risk groups. (**H**) Distribution of patients according to the median risk score. (**I**) The survival status distribution for each patient of a different risk group.

**Figure 5 cancers-14-05771-f005:**
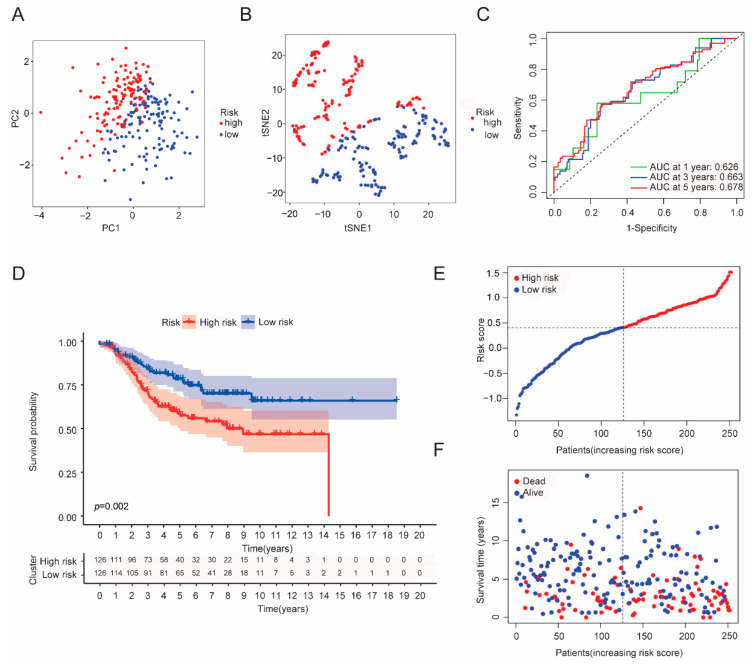
Validation of the risk score model in the validation set (GEO cohort). (**A**) PCA plot for BRCA patients. (**B**) tSNE plot for BRCA patients according to the risk score. (**C**) Time-dependent ROC curve analysis for BRCA patients. (**D**) Comparison of Kaplan–Meier curve analysis between two risk groups. (**E**) Distribution of patients in validation set based on the risk score. (**F**) The survival status distribution for each patient of different risk groups.

**Figure 6 cancers-14-05771-f006:**
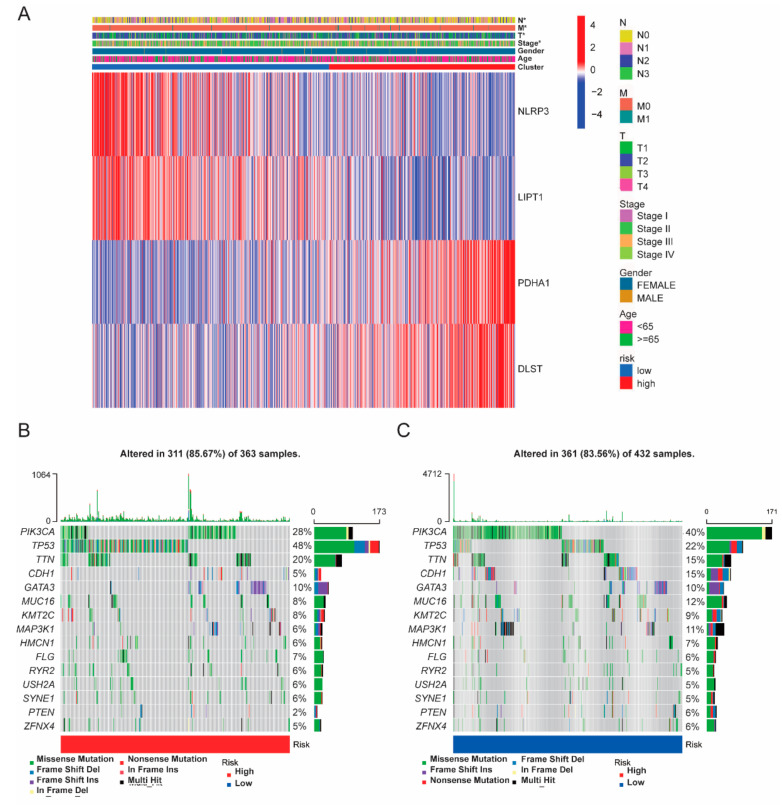
The cuproptosis-related DEGs and somatic mutation frequency rate among the breast cancer risk group. (**A**) Heatmap of the cuproptosis-related DEGs based on the breast cancer cluster. (**B**) Somatic mutation frequency rate based on the high-risk group. (**C**) Somatic mutation frequency rate based on the low-risk group.

**Figure 7 cancers-14-05771-f007:**
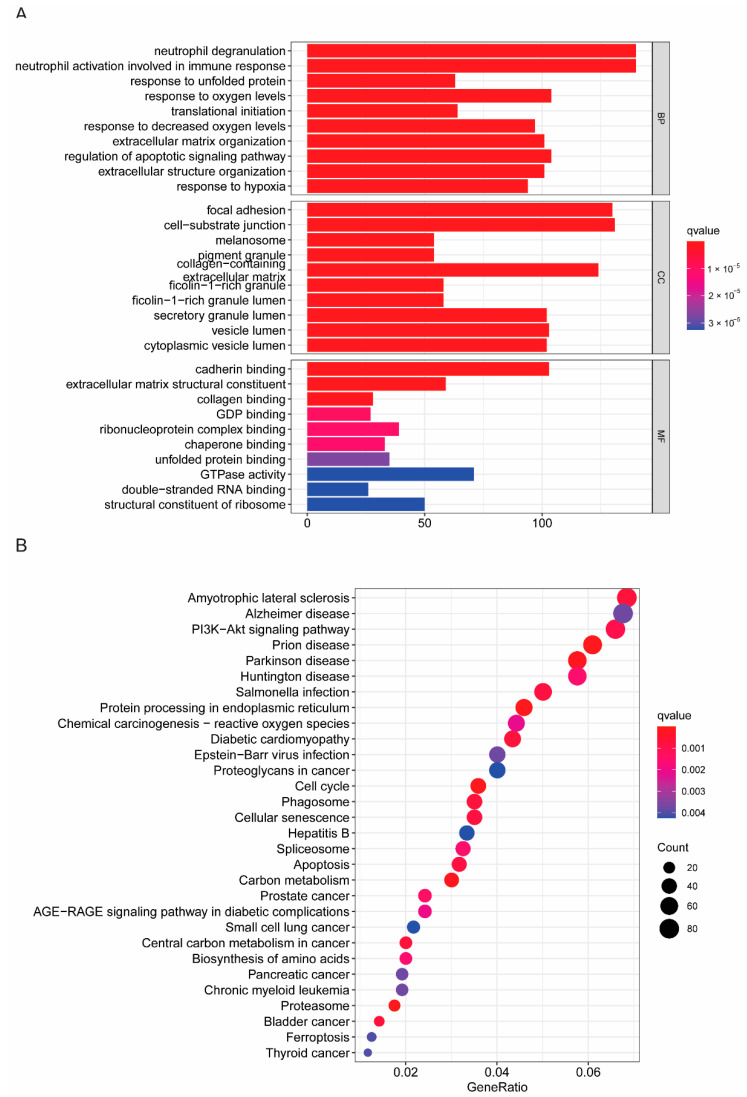
Molecular biological functional analysis based on the DEGs among the two-risk groups in the TCGA cohort. (**A**) Bar plot for KEGG pathways. (**B**) Bubble plot for GO enrichment.

**Figure 8 cancers-14-05771-f008:**
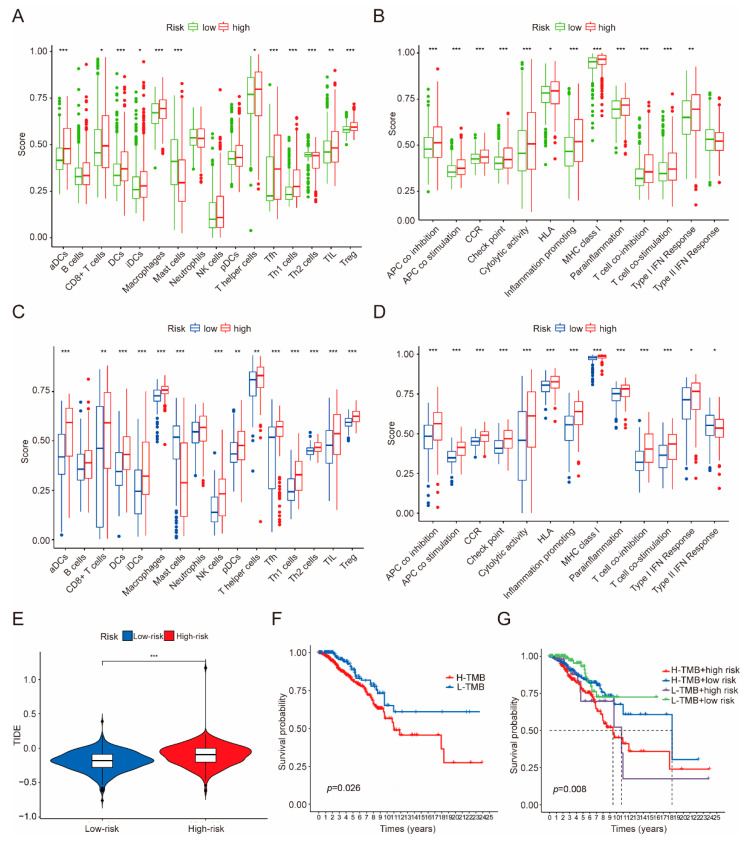
Comparison analysis of immune cells and immune pathways based on ssGSEA scores. (**A**,**B**) Comparison analysis of the enrichment scores of immune cells and immune-related pathways in training dataset (TCGA). (**C**,**D**) Comparison of the immune signaling pathways between low- and high-risk groups in validation dataset (GEO). The *p* values are shown as significant if * *p* < 0.05; ** *p* < 0.01; *** *p* < 0.001. (**E**) TIME comparison among risk groups in training set. (**F**) Kaplan–Meier curve analysis for comparison of the TMB between low- and high-risk groups. (**G**) Kaplan–Meier curve analysis for comparison of the low- and high-risk subgroups combined with TMB.

## Data Availability

The data in the paper can be found in the TCGA database (https://portal.gdc.cancer.gov/) and the GEO database (https://www.ncbi.nlm.nih.gov/geo/query/acc.cgi?acc=GSE21653), and all data is publicly available.

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
