# Peer review of "Prognostic Significance of Cuproptosis-Related Gene Signatures in Breast Cancer Based on Transcriptomic Data Analysis"

_cancers, 2022, doi:10.3390/cancers14235771_

Round 1

Reviewer 1 Report

cancers-1993329 Review

Prognostic significance of cuproptosis-related gene signatures in breast cancer based on transcriptomic data analysis

This paper examines the prognostic prediction of BRCA patients by “Cuprotosis” in breast cancer. However, it seems difficult to reveal heterogeneity only by the factor “Cuprotosis”.

Author Response

We would like to express our gratitude for the time and effort of each reviewer. We appreciate the reviewers’ comments concerning our manuscript entitled as “Prognostic significance of cuproptosis-related gene signatures in breast cancer based on transcriptomic data analysis”. We have tried our best to address these concerns by providing further evidence and explanation to support our data. Those comments are all valuable and helpful for us to revise and improve our manuscript. We believe that this has resulted in a significantly stronger manuscript. The corrections and the responses to each reviewer’s comments are listed as following:

Prognostic significance of cuproptosis-related gene signatures in breast cancer based on transcriptomic data analysis

This paper examines the prognostic prediction of BRCA patients by “Cuproptosis” in breast cancer. However, it seems difficult to reveal heterogeneity only by the factor “Cuprotosis”.

Response: We thank the reviewer for this comment and completely agree with him /her. Cancer is dynamic and become heterogenous during the course of carcinogenesis. Tumor heterogeneity can be observed in different aspects, including cellular morphology, metabolism, proliferation, angiogenic and metastatic potential and even the potential to evade immune attacks, leading to the production of subpopulations of cells1. Heterogeneity has been reported to show critical impacts on drug responses and patient prognoses2. Since tumor heterogeneity is determined by both heritable and non-heritable mechanisms3, cuproptosis, indeed, is impossible to be the only factor to reveal heterogeneity. However, in our study, we didn’t aim to reveal heterogeneity of breast cancer only by “Cuproptosis”-associated gene signature. The goal of this work is to add information on cancer behavior of specific class of patients, and to provide the proof-of-concept that cuproptosis signalling may be of great potential to be part of biomarker panel. In the future, once we can understand better different aspects of cancer cell behavior and the interactions among different cell types within tumor microenvironment, more sophisticated risk models could be able to be constructed for more accuracy of prediction. In addition, we have adjusted the descriptions in the relevant sentences and added this concern as a limitation in the discussion section (line 430-436).

Also, we have improved our introduction/ references/ methods/ results/ discussion and conclusion sections.

References:

  1. Fisher, R., Pusztai, L. & Swanton, C. Cancer heterogeneity: implications for targeted therapeutics. Br. J. Cancer 108, 479–485 (2013).
  2. Dagogo-Jack, I. & Shaw, A. T. Tumour heterogeneity and resistance to cancer therapies. Nat. Rev. Clin. Oncol. 15, 81–94 (2018).
  3. Marusyk, A. & Polyak, K. Tumor heterogeneity: causes and consequences. Biochim. Biophys. Acta 1805, 105–117 (2010).

Reviewer 2 Report

The study is well done, the material is large enough and the methods look reliable. However the study is based on extensive and very recent literature, gives some new information and this warrants its publication.

Author Response

We would like to express our gratitude for the time and effort of each reviewer. We appreciate the reviewers’ comments concerning our manuscript entitled as “Prognostic significance of cuproptosis-related gene signatures in breast cancer based on transcriptomic data analysis”. We have tried our best to address these concerns by providing further evidence and explanation to support our data. Those comments are all valuable and helpful for us to revise and improve our manuscript. We believe that this has resulted in a significantly stronger manuscript. The corrections and the responses to each reviewer’s comments are listed as following:

The study is well done, the material is large enough and the methods look reliable. However, the study is based on extensive and very recent literature, gives some new information and this warrants its publication.

Response: We do appreciate the reviewer’s interests. Based on reviewer’s comments, we did more analyses and added more information in the result/ discussion sections of our new version of manuscript, including Figure S3 (Comparison of the AUC between our risk model and other gene models) and Figure S8 (Comparison of AUC and DCA between our nomogram model and TNM model). We believe, in this way, our study could be more reliable. In addition, we polished our language as well. Hopefully the reviewer will be satisfied with our revision.

Reviewer 3 Report

The authors have presented a very clear and interesting work on potential prognostic markers of breast cancer using cuproptosis-related signatures. They have explained their methods and conclusions succinctly and clearly. Having said that the following questions need to be answered:

1) How does this approach compare or contrast with Li, Jia, et al. "The cuproptosis-related signature predicts prognosis and indicates immune microenvironment in breast cancer." Frontiers in genetics 13 (2022).

2) The authors need to provide how their AUCs compare with other approaches in literature that cite different cuproptosis related gene signatures? This will also help propose optimal therapeutic combination targets

3) In fig 4. How did the authors distinguish between the high and low risk groups. In both the PCA and TSNE plots they dont seem to be different at all.

4) In general the plots and the inset data need to be more legible for the readers to understand better

Author Response

We would like to express our gratitude for the time and effort of each reviewer. We appreciate the reviewers’ comments concerning our manuscript entitled as “Prognostic significance of cuproptosis-related gene signatures in breast cancer based on transcriptomic data analysis”. We have tried our best to address these concerns by providing further evidence and explanation to support our data. Those comments are all valuable and helpful for us to revise and improve our manuscript. We believe that this has resulted in a significantly stronger manuscript. The corrections and the responses to each reviewer’s comments are listed as following:

The authors have presented a very clear and interesting work on potential prognostic markers of breast cancer using cuproptosis-related signatures. They have explained their methods and conclusions succinctly and clearly. Having said that the following questions need to be answered:

1) How does this approach compare or contrast with Li, Jia, et al. "The cuproptosis-related signature predicts prognosis and indicates immune microenvironment in breast cancer." Frontiers in genetics 13 (2022).

Response: We thank the reviewer for pointing out the question. Indeed, Jia Li et al. reported a nice model to efficiently predict prognosis and even response to immunotherapy. There are two major differences between our model and the model established by Jia Li et al. Firstly, the datasets used to build risk models are different. In our study, we used the whole samples from TCGA-BRCA dataset to train the risk model. However, in the work reported by Jia Li et al., the training cohort was derived from half of TCGA-BRCA dataset (the whole TCGA-BRCA dataset was divided into two cohorts at a 1:1 ratio). Secondly, their model was built based on 13 differentially-expressed cuproptosis regulators (between breast cancer and normal samples) selected from the original report firstly discovering cuproptosis1. Regarding our model, we have applied expanded cuproptosis-related DEGs (18 DEGs) according to different studies1-5. Collectively, we believe both models are great, but established based on different datasets and cuproptosis-related regulators. We also added this information in our discussion section (line 422-430).

References:

  1. Tsvetkov, P.; Coy, S.; Petrova, B.; Dreishpoon, M.; Verma, A.; Abdusamad, M.; Rossen, J.; Joesch-Cohen, L.; Humeidi, R.; Spangler, R.D.; et al. Copper induces cell death by targeting lipoylated TCA cycle proteins. Science 2022, 375, 1254-1261, doi:10.1126/science.abf0529.
  2. Wang, Y.; Zhang, L.; Zhou, F. Cuproptosis: a new form of programmed cell death. Cell Mol Immunol 2022, 19, 867-868, doi:10.1038/s41423-022-00866-1.
  3. Bian, Z.; Fan, R.; Xie, L. A Novel Cuproptosis-Related Prognostic Gene Signature and Validation of Differential Expression in Clear Cell Renal Cell Carcinoma. Genes (Basel) 2022, 13, doi:10.3390/genes13050851.
  4. Kahlson, M.A.; Dixon, S.J. Copper-induced cell death. Science 2022, 375, 1231-1232, doi:10.1126/science.abo3959.
  5. Tang, D.; Chen, X.; Kroemer, G. Cuproptosis: a copper-triggered modality of mitochondrial cell death. Cell Res 2022, 32, 417-418, doi:10.1038/s41422-022-00653-7.

2) The authors need to provide how their AUCs compare with other approaches in literature that cite different cuproptosis related gene signatures? This will also help propose optimal therapeutic combination targets.

Response: We thank the reviewer for raising the concerns here. Indeed, according to reviewer’s useful comments, we compare our AUCs with some other approaches in Figure S3 of new version of manuscript (line 219-221) (please check the attachment). Our results showed the AUC value of our gene model our risk model was higher than that of others1, 2.

References:

  1. Li, W.; Zhang, X.; Chen, Y.; Pang, D. Identification of cuproptosis-related patterns and construction of a scoring system for predicting prognosis, tumor microenvironment-infiltration characteristics, and immunotherapy efficacy in breast cancer. Front Oncol 2022, 12, 966511, doi:10.3389/fonc.2022.966511.
  2. Li, J.; Wu, F.; Li, C.; Sun, S.; Feng, C.; Wu, H.; Chen, X.; Wang, W.; Zhang, Y.; Liu, M.; et al. The cuproptosis-related signature predicts prognosis and indicates immune microenvironment in breast cancer. Front Genet 2022, 13, 977322, doi:10.3389/fgene.2022.977322.

3) In fig 4. How did the authors distinguish between the high and low risk groups. In both the PCA and TSNE plots they dont seem to be different at all.

Response: We thank the reviewer for pointing out the concern and sorry to make this so confusing. Indeed, the risk models was built based on high-/ low- CRGs expression level and visualized by TSNE and PCA analyses. Of course, we have to admit that we could not separate the two subpopulations completely potentially due to the large sample size. But still our results demonstrated that we could observe a relatively higher significant difference between two groups regarding PCA plot ( PC1(minimal: -5.30754, maximum: 5.65042) and PC2(minimal:-3.84088, maximum: 3.32266)). Also, our data suggested low- and high- risk models showed significant difference regarding prognosis and our risk model showed high AUC value in predicting prognosis. Moreover, we found other published studies also have this situation. Even though they also presented partially overlapping between two subtypes, the models built were also shown to efficiently predict clinical outcomes (Reference 11 Figure 3D; Reference 22 Figure 3F). Therefore, we believe our model is still a useful tool and the subpopulations could be divided effectively to some extent. Thank the reviewer again for these useful comments and hope our answer could release your concerns.

References:

  1. SONG, S., ZHANG, M., XIE, P., WANG, S. & WANG, Y. (2022), "Comprehensive analysis of cuproptosis-related genes and tumor microenvironment infiltration characterization in breast cancer", Front Immunol, Vol. 13978909.
  2. LI, J., WU, F., LI, C., SUN, S., FENG, C., WU, H., CHEN, X., WANG, W., ZHANG, Y., LIU, M., LIU, X., CAI, Y., JIA, Y., QIAO, H., ZHANG, Y. & ZHANG, S. (2022), "The cuproptosis-related signature predicts prognosis and indicates immune microenvironment in breast cancer", Front Genet, Vol. 13977322.

4) In general, the plots and the inset data need to be more legible for the readers to understand better.

Response: We are grateful for this suggestion. We have readjusted all the plots and the inset data with high resolution, which hopefully become more legible for the readers to understand better.

Reviewer 4 Report

The paper is associated with specific molecular cancer subtypes. The overall methodology used by authors seems correct. Of course, the cuproptosis cannot be the only factor characterizing specific cancer subtypes, but it adds information on cancer behavior of specific class of patients. Moreover, the predictive value of CRGs in immune score in tumor microenvironment and patient prognosis in breast cancer have been reported, and different statistic tests were done to assess this hypothesis.

Although many studies are needed to support the assumption of authors of managing cancer by regulating the level of copper in the body, this is another point of view of nutrigenomic studies that can add a piece of evidence in the comprehension of molecular mechanisms of BC.

Results and conclusion sections could be improved, could be clearer in the exposition.

Author Response

We would like to express our gratitude for the time and effort of each reviewer. We appreciate the reviewers’ comments concerning our manuscript entitled as “Prognostic significance of cuproptosis-related gene signatures in breast cancer based on transcriptomic data analysis”. We have tried our best to address these concerns by providing further evidence and explanation to support our data. Those comments are all valuable and helpful for us to revise and improve our manuscript. We believe that this has resulted in a significantly stronger manuscript. The corrections and the responses to each reviewer’s comments are listed as following:

The paper is associated with specific molecular cancer subtypes. The overall methodology used by authors seems correct. Of course, the cuproptosis cannot be the only factor characterizing specific cancer subtypes, but it adds information on cancer behavior of specific class of patients. Moreover, the predictive value of CRGs in immune score in tumor microenvironment and patient prognosis in breast cancer have been reported, and different statistic tests were done to assess this hypothesis.

Although many studies are needed to support the assumption of authors of managing cancer by regulating the level of copper in the body, this is another point of view of nutrigenomic studies that can add a piece of evidence in the comprehension of molecular mechanisms of BC.

Results and conclusion sections could be improved, could be clearer in the exposition.

Response: Thanks for reviewer’s support and helpful comments. Indeed, as reviewer mentioned, it is of great challenge and impossible to use the only factor “cuproptosis” to class the subtypes of cancer patients. The aim of our study is to provide the proof-of-concept that cuproptosis signalling may be of great potential to be part of biomarker panel. Hopefully, our work could also attract the attention of other researchers to further focus on the role of cuproptosis in the potential clinical application of breast cancer field. According to reviewer’s suggestion, we have revised results and conclusion sections in our new version of manuscript.

Round 2

Reviewer 1 Report

cancers-1993329 Review

Prognostic significance of cuproptosis-related gene signatures in breast cancer based on transcriptomic data analysis

This paper has been improved in accordance with the instructions. I consider this manuscript acceptable.